# Prevalence and Predictors of Using Antibiotics without a Prescription in a Pediatric Population in the United States

**DOI:** 10.3390/antibiotics12030491

**Published:** 2023-03-01

**Authors:** Kiara Olmeda, Barbara W. Trautner, Lindsey Laytner, Juanita Salinas, Stephanie Marton, Larissa Grigoryan

**Affiliations:** 1Department of Family and Community Medicine, Baylor College of Medicine, Houston, TX 77098, USA; 2Department of Medicine, Section of Health Services Research, Baylor College of Medicine, Houston, TX 77030, USA; 3Center for Innovations in Quality, Effectiveness, and Safety (IQuESt), Michael E. DeBakey Veterans Affairs Medical Center, Houston, TX 77021, USA; 4Department of Pediatrics, Baylor College of Medicine, Houston, TX 77030, USA; 5Texas Children’s Health Plan, Houston, TX 77067, USA

**Keywords:** antimicrobial resistance, pediatric, antibiotic use, non-prescription antibiotic use, antimicrobial stewardship

## Abstract

Non-prescription antibiotic use (using antibiotics without clinical guidance) increases the risk of the development of antibiotic resistance, adverse drug reactions, and other potential patient harm. Few studies have explored non-prescription use in children in the U.S. From January 2021 to April 2022, a diverse sample of caregivers of children under 18 years were surveyed in English and Spanish at two safety net clinics in Texas. We assessed the prevalence of antibiotic use in children in the previous 12 months, storage of antimicrobials, and intended use of non-prescription antibiotics (professed intention for future non-prescription antibiotic use). We also measured sociodemographic factors, types of antibiotics used, and symptoms that trigger non-prescription use. The response rate was 82%, and 17% were surveyed in Spanish. Of 322 participants surveyed, three Spanish-speaking caregivers reported giving non-prescription antibiotics to their child in the previous 12 months. Approximately 21% (*n* = 69) reported storing antimicrobials at home, specifically amoxicillin (*n* = 52), clindamycin (*n* = 10), cephalexin (*n = 5*), penicillin (*n* = 3), and trimethoprim/sulfamethoxazole (*n = 3*). Nearly 15% (*n* = 46) reported intention to give non-prescription antibiotics to their children. Younger caregiver age was associated with storage and intended use of non-prescription antibiotics. Our findings will guide the development of an educational intervention to decrease non-prescription antibiotic use.

## 1. Introduction

Antibiotic misuse and antimicrobial resistance are global concerns, particularly among pediatric populations [1,2,3,4,5]. The burden of antimicrobial resistance in the United States remains high, with more than 2.8 million antibiotic-resistant infections and over 35,000 deaths each year [1,2]. The World Health Organization (WHO) estimates that antibiotic resistance causes at least 700,000 deaths across all ages each year worldwide, with 200,000 of these deaths occurring in children [4,5]. Pediatric populations are particularly vulnerable as they are more susceptible to certain infections, often receive more antibiotics than adults, and may experience prolonged illnesses or complications because of antibiotic resistance [5,6]. Antibiotic use in children has also been potentially associated with an increased risk of asthma, obesity, juvenile idiopathic arthritis, and reduced microbiome diversity [6,7,8,9]. Non-prescription antibiotic use (non-prescription use) is the practice of obtaining and taking antibiotics without consulting a medical professional, such as purchasing from stores or flea markets in the United States (U.S.) or abroad, taking another person’s antibiotics, or using stored antibiotics for an indication other than the one for which the antibiotics were originally prescribed [10,11]. Non-prescription antibiotic use among children is of great concern as it may involve extremely short courses, incorrect drug and dosage choices, and unnecessary therapy [11,12]. All of these practices can in turn foster the emergence of resistant organisms. Previous studies have shown that non-prescription use in children is prevalent in low- and middle-income countries [13,14,15,16]. For example, a 2019 study conducted in Lima, Peru, uncovered that nearly one in four parents surveyed reported giving their child antibiotics without a prescription in the past year and that many of these parents did not understand the risks associated with this practice [15]. Non-prescription antibiotic use was widespread among parents of low socioeconomic status from east Asia, south Asia, and sub-Saharan African countries [13,14], and antibiotics were often used to treat viral upper respiratory tract infections [15,16].

In the U.S., a 2019 scoping review on non-prescription use found that very few studies of this topic have been performed in pediatric populations or their caregivers [10]. Among the three such studies identified, the prevalence rates of non-prescription use ranged from 3% to as high as 48% [10,17,18]. One study completed in Wisconsin and Minnesota found that 3% of parents of young children reported that they had used antibiotics for themselves or for their child without consulting a clinician in the previous 6 months [17]. The highest estimate of 48% came from a 2018 national Internet-based survey of parents about pediatric antibiotic diversion; specifically, of the 48% (*n =* 219) who kept leftover antibiotics, 73% (*n =* 153) subsequently diverted those antibiotics to their child’s siblings, unrelated children, and unrelated adults [18]. Our study aimed to estimate the prevalence rate of non-prescription antibiotic use among children and to identify associated predictors from a racially and ethnically diverse sample of parents and caregivers. Very few pediatric studies have been conducted in the U.S. outpatient setting [17,18,19,20]. Of the four such studies that we identified, only two were conducted in two languages (English and Spanish) [19,20]. Only one recently completed study was in the southwest region of the United States [20]. A prior investigation in adults identified underserved populations as more likely to use non-prescription antibiotics [11]. For this reason, we sought to conduct our pediatric survey in clinics specifically caring for underserved families.

Our study seeks to (1) estimate the prevalence of prior non-prescription antibiotic use, storage of antimicrobials, and intended use in a racially and ethnically diverse sample of parents and caregivers of children under 18 years old receiving care at two Texas Children’s Health Plan Clinics [21,22] in a large, urban area; (2) examine the types of antimicrobials stored, duration of use, and the symptoms that led parents to give antibiotics without a prescription to their children; and (3) identify key sociodemographic factors associated with storage and intended use of non-prescription antibiotics.

## 2. Results

### 2.1. Population Demographics, Survey Response Rate, and Analysis

We screened 425 participants in Houston, Texas, from January 2021 to April 2022. Of those screened, 400 participants were interested in responding to our survey (See Appendix A). A total of 392 parents met the eligibility criteria, and 322 consented to participate and completed the survey (response rate, 82%). Data analysis was completed on responses from a total of 322 participants. There were 79 surveys completed via phone due to the ongoing COVID-19 pandemic and 243 surveys completed in person.

In addition to inquiring about parent behaviors related to antibiotic use, this survey also captured participants’ sociodemographic characteristics in Table 1. Participants were primarily Hispanic/Latino (51%) or African American or Black (48%), and female (94%). Many parents/caregivers were aged between 31 and 40 years (49%), followed by 18 and 30 years (32%), and 41 and 69 years (18%) (Table 1). All parents interviewed had a child or children insured by the Texas Children’s Health Plan [21,22].

#### 2.1.1. Non-Prescription Antibiotic Use, Storage of Antimicrobials, and Intended Use

A total of 190 participants reported that their child had taken antibiotics in the previous 12 months, but only three participants reported that this antibiotic use had been outside the context of a healthcare prescribing situation (non-prescription use) (Figure 1). These three participants said that they had obtained antibiotics without a prescription from a local store or market—such as flea markets, yerberias (variety stores selling Central and South American products including medicines), and herbalists shops—in the U.S. In total, 69 caregivers reported having stored antimicrobials at home, excluding any child’s currently prescribed antibiotics courses, and 46 (14%) participants said that they would give non-prescription antibiotics to their child without contacting a healthcare professional. There were 18 (6%) caregivers who had stored antimicrobials at home and intended to use non-prescription antibiotics (Figure 1). The primary source for caregivers who reported having stored antimicrobials at home or intended to use them was left over from a previous prescription. 

#### 2.1.2. Reported Symptoms/Illnesses and Types of Antibiotics for Non-Prescription Use

Ear infection or ear pain, as well as respiratory symptoms/illnesses (including bronchitis, sinus infection, and sore throat), were the most common reasons for intended use of non-prescription antibiotics, followed by urinary tract infection (UTI) or cystitis, fever, and runny nose or cold (Figure 2).

#### 2.1.3. Types of Antimicrobials Stored at Home

Amoxicillin was the most commonly stored antibiotic, followed by clindamycin and cephalexin (Figure 3). The duration of non-prescription use ranged from 1 to 30 days (median 7 days).

#### 2.1.4. Univariate Analysis

Table 2 shows the results of the univariate logistic regression analysis on factors associated with the following outcome variables: (1) storage of antimicrobials, (2) intended use of non-prescription antibiotics, and (3) storage and intended use. Candidate sociodemographic factors associated with storage and intention to use included age, number of children in household, race and ethnicity, and education. We had a high proportion of respondents who did not know or preferred not to report their household income (23%); therefore, we excluded this variable from the regression analyses. Candidate predictors (*p* < 0.20) from each outcome variable are presented in Table 2. The third outcome variable, storage and intended use, identified the highest-risk participant sub-group for future non-prescription use—those who reported storing antimicrobial medications at home and intending to use them without a prescription. Older caregivers and those with a higher number of children in the household were less likely to store antimicrobials. Older caregivers were also less likely to report intended use of non-prescription antibiotics for their children.

## 3. Discussion

We found that 21% of our surveyed caregivers had oral antimicrobials readily available at home, and 14% indicated that they would give antibiotics to their children without consulting a healthcare provider. The combined variable of storage and intended use identified 18 (6%) caregivers as the highest-risk participant sub-group for future non-prescription use—those who reported storing antimicrobial medications at home and intending to use them without a prescription. Storage of antimicrobials and intended use of non-prescription antibiotics were similar across all studied racial and ethnic groups. Caregivers reported that many antibiotics stored in their household were left over from a previous prescription. This finding suggests noncompliance with the recommended duration of therapy. The other source that caregivers cited that they obtained antibiotics without a prescription was in a store or market in the U.S., indicating that U.S. residents are able to purchase antibiotics without a prescription for their children [20,23]. The common symptoms/illnesses driving intended use of non-prescription antibiotics are mostly viral respiratory infections. Younger caregivers were more likely to store antibiotics or report the intention to use them without a prescription, with the strongest association in the 18- to 30-year-old group.

The prevalence rate of non-prescription antibiotic use in children in our study is similar to that reported in another face-to-face survey, but lower than the rates reported in a US-based Internet study [17,18,19]. In a study conducted in Wisconsin and Minnesota using random digit dialing, nine parents (3%) reported that they had used antibiotics for themselves or their children in the past 6 months without first consulting a doctor [17]. On the other hand, of the 454 parents who completed an Internet-based survey, 219 parents (48%) reported saving antibiotics instead of disposing of them and 73% (*n* = 153) subsequently gave those antibiotics to their child’s siblings, unrelated children, and unrelated adults [18]. In another face-to-face survey of 396 caregivers of children, 65 (16%) had stored antibiotics at home and 12% reported past or intended non-prescription use [19]. Similar to our findings, the researchers found no significant association between any racial or ethnic group and non-prescription use [19]. The differences observed between these studies can be explained by heterogeneous research methods, including different sampling techniques, study settings, and timeframe [17,18,19]. 

Another possible explanation for the lower prevalence of previous non-prescription use in our study is that our survey was performed during the COVID-19 pandemic when the antibiotic prescribing volume was lower than it was pre-pandemic [24]. This phenomenon likely reflects decreased transmission of non-COVID-19 infections secondary to non-pharmaceutical interventions (e.g., masking, social distancing, and school closures) [24]. During the pandemic, pediatric caregivers were hesitant to seek healthcare in general and may not have had the opportunity to obtain an antibiotic prescription [24,25]. The evidence to support this comes from an analysis of over 2 million children enrolled per year in a U.S. insurance claims database [26]. Researchers identified striking changes in healthcare utilization by children during the COVID-19 pandemic; more specifically, utilization decreased across all sites of care but was more pronounced for sick visits (at PCPs, urgent cares, and EDs) than for well-child visits [26]. 

Our study has certain limitations. First, the survey relies on self-reported data, which may have underestimated the true prevalence rate of non-prescription use due to social desirability bias. Participants may be more likely to deny giving antibiotics without a prescription to their children, especially if they are interviewed in a healthcare setting and are aware that this behavior is inappropriate [11]. To control for any social desirability bias, our survey questions were formulated neutrally to ensure that the source of the non-prescribed antibiotic could be chosen from six predefined sources or ‘Other source.’ Second, our surveys were only performed in English and Spanish and not in other languages. However, English and Spanish are the two most common languages in our geographic area and in the United States. Third, our study was conducted in a multicultural urban area, which may not represent the overall U.S. population. The last limitation stems from recruiting during the pandemic, as any child exhibiting symptoms potentially consistent with COVID-19 was prohibited from clinic waiting rooms and was required to seek care via telemedicine visits or immediately ushered to an exam room.

In a recent systematic review, the prevalence rate of antibiotic self-medication among children was highest in studies conducted in the Middle East (34%), while the lowest prevalence was found in Europe (8%) [27]. An additional finding from this systematic review was that only two studies, one conducted in Tanzania and the other in Jordan, reported parent age as statistically significant, but the results were mixed [27]. In the first study completed in Tanzania, researchers found that parents younger than 40 tended to administer more antibiotics without consulting a physician [27]. On the contrary, in a study from Jordan, parents older than 40 were more inclined to self-medicate their children with antibiotics [27]. However, our findings align with previous studies in the U.S. that identified younger adults as having a higher propensity to use non-prescribed antibiotics [11,28]. Further research is needed to understand why younger adults in the U.S. are more likely to use non-prescription antibiotics and to target them in future community stewardship interventions [29,30,31,32]. Qualitative research methods may help identify gaps in knowledge and address misconceptions parents may have about antibiotics for their children [33,34].

## 4. Materials and Methods

### 4.1. Study Setting, Sample Size, and Recruitment

This study consisted of a brief survey comprising 15 questions conducted by a research coordinator who is fluent in English and Spanish between January 2021 and April 2022 (See Appendix A). Recruitment occurred in two Texas Children’s Health Plan (TCHP) clinics, which are pediatric-safety net clinics serving families whose maximum annual income meets eligibility, under $36,996 for Medicaid and $56,055 for a family of four for CHIP (Children’s Health Insurance Program) [21,22]. TCHP provides medical coverage to kids, teens, pregnant women, and adults through CHIP and Medicaid. Through this plan, families have access to a large network of doctors and resources throughout Texas, including Harris County [21,22]. Within these two clinics, we utilized a race and ethnicity-stratified systematic random sampling to select our sample of adult parents and caregivers. In our previous study that included two safety-net clinics, 30% of the respondents reported that they would use non-prescription antibiotics [11]. If the maximum expected prevalence is 30%, to obtain a precision of 0.05, the sample size needed is 323 [11]. The clinic and research staff gave a flyer to every third patient who checked in for a primary care visit. The recruitment flyer introduced the study and asked the patient to approach the study research coordinator if interested in participating. The research coordinator then invited willing participants to complete a short questionnaire about their use of antibiotics. Exclusion criteria were parents and caregivers less than 18 years old and an inability or unwillingness to complete the survey. All eligible participants were adult parents or caregivers of children under 18 years old.

All parents or caregivers were screened by a bilingual research coordinator using the following questions: (1) Does your child or children receive care at either of the two Texas Children’s Health Plan Clinics? (2) Do you and the child or children under your care live in the same household? If the caregiver answered yes to these questions, then the caregiver was invited to participate in the anonymous survey. To reduce recall errors, the back of the recruitment flyer that parents received provided the photos and names of the most common antibiotics prescribed for children to use as a reference (see Appendix A). All participants received $15 in compensation for their time. The research coordinator was present in the waiting areas to conduct the survey in person when permitted during the COVID-19 pandemic. Throughout the months of January to December of 2021, the coordinator had limited time in waiting areas, and interviews were performed by phone.

### 4.2. Survey Design

This survey was adapted from a pretested and standardized questionnaire on the prevalence of self-medication using antibiotics in Europe and the U.S. designed for adult participants [11,35,36,37,38,39,40]. The survey was reviewed and revised by a pediatrician (S.M.) with input from two pediatric clinic directors about the acceptability of survey questions for pediatric caregivers. Bilingual research coordinators (K.O. and J.S.) translated the revised survey into Spanish, and a back-translation strategy was used on the Spanish version to check for semantic equivalence in both English and Spanish surveys [40]. Our study included a population of parents and caregivers that are racially and ethnically diverse. By conducting our survey in both English and Spanish, we were able to capture the responses of Spanish-speaking participants who are underrepresented in surveys in the United States [41]. Our survey included questions examining a range of potential sources and possible predictors for using antibiotics without a prescription.

Surveys were completed in approximately 20 min, and the English version of the full survey is listed in Appendix A. The survey queried the participants on their child’s use of antibiotics during the last 12 months, the name of the antibiotics used, the duration of use, and how the antibiotics were obtained [11,35]. Questions also asked about whether antibiotics were stored at home and whether participants would consider using antibiotics without contacting a healthcare professional [11,35]. 

### 4.3. Variables of Interest

Non-prescription antibiotic use (non-prescription use) was defined as giving antibiotics to a child without consulting a medical professional, such as purchasing from stores or flea markets in the United States (U.S.) or abroad, using another person’s antibiotics, or using stored antibiotics for an indication other than the one for which the antibiotics were originally prescribed [10,11]. Participants were classified as non-prescription antibiotic users if they reported giving their child or children antibiotics without a prescription in the previous 12 months. To assess the storage of antimicrobials, all participants who reported storing antimicrobial medicines at home were included. Intended use of non-prescription antibiotics is defined as a parent or caregiver who answered “yes” or “maybe” to the question: In general, would you give antibiotics to your child without contacting a doctor/nurse/dentist/hospital? We also included in the analysis those who reported both storing antimicrobial medications at home and intending to use them without a prescription, as this subgroup may be at highest risk for non-prescription use. 

### 4.4. Statistical Analysis

We performed descriptive statistics on all study variables. Descriptive statistics were used to estimate the prevalence rates for the storage of antimicrobials, intention to use non-prescription antibiotics, and storage coupled with intention to use. We used descriptive statistics to analyze the frequency of symptoms that trigger non-prescription use, the specific antibiotics used, and the duration of antibiotic use. The associations between caregiver characteristics were studied with logistic regression analyses using three outcome variables: (1) storage of antimicrobials, (2) intention to use antibiotics without a prescription, and (3) storage of antimicrobials and intention to use. Data analysis was conducted using SPSS software version 28.0.1.1 (14). 

## 5. Conclusions

Our findings demonstrate that greater efforts to reduce the use of antibiotics without a prescription in pediatric populations are needed. There was very low self-reported non-prescription antibiotic use by pediatric parents and caregivers, and all three respondents who acknowledged this activity were Spanish-speaking. However, we found 18 (6%) at very high risk for non-prescription use, on account of both possessing stored antibiotics and having an intention to give them to their children. An identified participant characteristic, such as young parent age, can inform the development of targeted interventions for specific age groups of parents to address gaps in knowledge and misconceptions contributing to inappropriate use of antibiotics. Future research should focus on the reasons why younger adults are more likely to use non-prescribed antibiotics and identify patient, healthcare system, and clinical encounter factors driving non-prescription use. Mixed methods studies exploring these factors within a conceptual framework can inform the development of future interventions to reduce this unsafe practice. Our findings highlight the need for antibiotic stewardship educational intervention on appropriate antibiotic use for parents and caregivers of children. Identifying the prevalence of non-prescription use in pediatric populations is the first step in developing appropriate antibiotic stewardship interventions to preserve the effectiveness of these drugs and reduce the emergence of antibiotic-resistant bacteria.

## Figures and Tables

**Figure 1 antibiotics-12-00491-f001:**
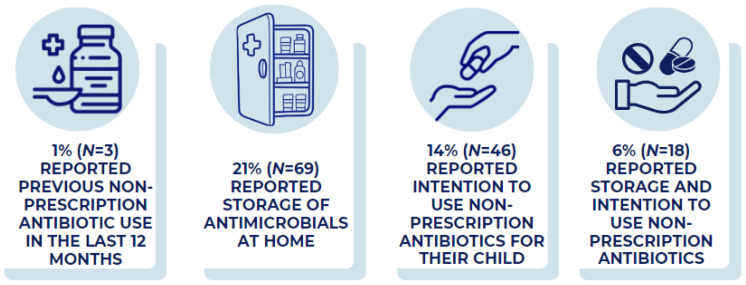
Prevalence of reported prior use, storage of antimicrobials, and intended use of antibiotics. Percentages represent participants who reported previous use of non-prescription antibiotics, storing antimicrobials at home, and intending to use non-prescription antibiotics for their child. See Methods for detailed definitions.

**Figure 2 antibiotics-12-00491-f002:**
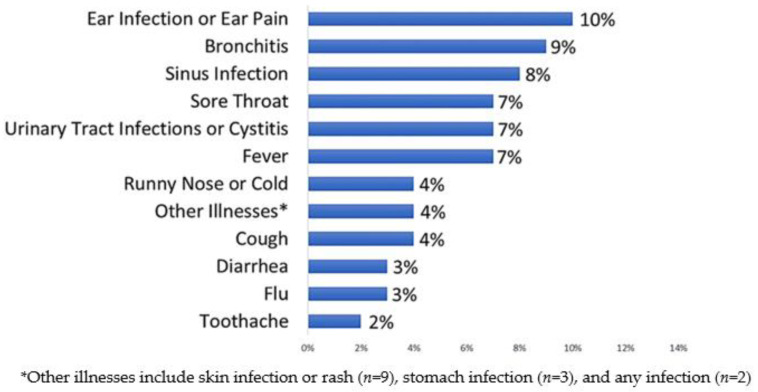
Prevalence of intended use per predefined symptom/illness. Percentages represent participants who reported intention to use non-prescription antibiotics for a given symptom/illness.

**Figure 3 antibiotics-12-00491-f003:**
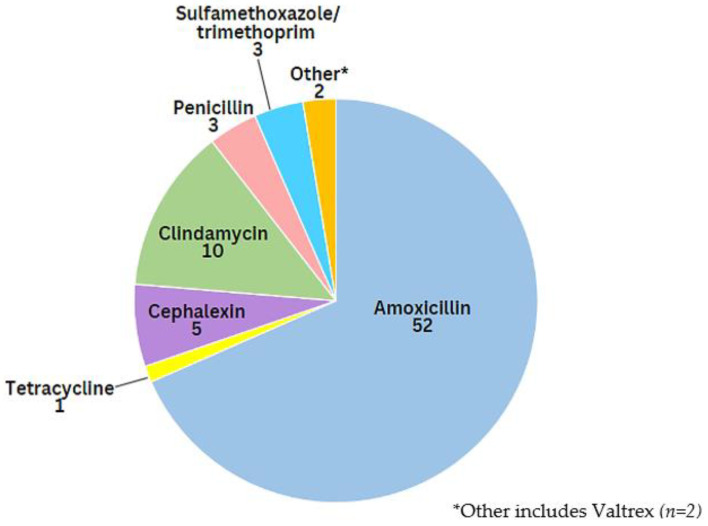
Types and number (*n*) of reported antimicrobials stored at home.

**Table 1 antibiotics-12-00491-t001:** Sociodemographic characteristics of participants (*n* = 322).

Characteristic	Value
Caregiver Age Group	34 (18–69)
18–30 years old	104 (32)
31–40 years old	159 (49)
41–69 years old	59 (18)
No. (%) of female participants	303 (94)
No. (%) of questionnaires completed in Spanish	54 (17)
No. (%) of participants of race/ethnicity	
Hispanic/Latino	164 (51)
African American or Black	144 (48)
Non-Hispanic White	8 (3)
Other ^a^	6 (2)
Texas Children’s Clinic location	
Greenspoint	162 (50)
Southwest	160 (50)
No. (%) of participants with education level	
High school or less	143 (44)
Some college or above	179 (56)
Number of children in household	
1–2 children	167 (52)
3–5 children	155 (48)
No. (%) of participants with household income	
<$20,000	101 (31)
≥$20,000 but <$40,000	102 (32)
≥$40,000 but <$60,000	34 (11)
≥$60,000 but <$100,000	6 (2)
Greater than $100,000 Don’t know/prefer not to say	4 (1) 75 (23)

^a^ Other includes Asian (*n* = 1), Pacific Islander/Hawaiian (*n* = 1), multiracial (*n* = 3), and declined (*n* = 1).

**Table 2 antibiotics-12-00491-t002:** Univariate logistic regression of predictors of stored antimicrobials and intended use of non-prescription antibiotics (*n* = 322).

	Stored Antimicrobials		Intended Use		Stored Antimicrobials and Intended Use	
Predictor	OR (95% CI)	*p*-Value ^b^	OR (95% CI)	*p*-Value ^b^	OR (95% CI)	*p*-Value ^b^
**Caregiver Age Group** 18–30 years old 31–40 years old 41–69 years old	0.94 (0.9–0.98) 1 (reference) 0.20 (0.11–0.38) (0.23–0.98)	**0.001** 0.47 **0.04**	0.95 (0.91–0.99) 1 (reference) 0.48 (0.24–0.94) 0.42 (0.16–1.11)	**0.04** **0.03** 0.08	0.89 (0.82–0.96) 1 (reference) 0.082 (0.02–0.37) 0.226 (0.05–1.03)	**0.004** **0.001** 0.06
**Number of children in household**						
1–2 children	1 (reference)		1 (reference)		1 (reference)	
3–5 children	0.33 (0.18–0.59)	**<0.001**	1.64 (0.87–3.09)	0.12	1.082 (0.42–2.80)	0.87
**Race and Ethnicity** Hispanic or Latino African American or Black White/Other ^a^						
	1 (reference)		1 (reference)		1 (reference)	
	1.49 (0.86–2.57)	0.15	1.17 (0.62–2.18)	0.63	1.86 (0.70–4.92)	0.21
	1.22 (0.32–4.64)	0.77	–			–
**Education** High school or less Some college or above						
	1 (reference) 1.42 (0.82–2.46)	0.21	1 (reference) 0.57 (0.30–1.06)	0.08	1 (reference) 0.49 (0.18–1.29)	0.15

^a^ Other includes Asian (*n* = 1), Pacific islander/Hawaiian (*n* = 1), multiracial (*n* = 3), declined (*n* = 1); ^b^ Bolded values represent a statistical significance of *p*-value < 0.05.

## Data Availability

The questionnaire and list of commonly prescribed antibiotics for children used in this study are provided in Appendix A.

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
