# Peer review of "Prevalence and Predictors of Using Antibiotics without a Prescription in a Pediatric Population in the United States"

_antibiotics, 2023, doi:10.3390/antibiotics12030491_

Round 1

Reviewer 1 Report

This is an interesting study on the administration of non-prescribed antibiotics to children in the US. It is an issue of crucial importance for public health since the over-consumption of antibiotics is directly connected to antimicrobial resistance which threatens to limit our treatment options. The article is well written and the results well founded and presented. A minor comment for the authors to add some more information about antimicrobial resistance in the US for the average reader. I found also intriguing the fact that there is access to antibiotics in drug stores in US without prescription since the system is well organized. The authors also refer that many participants stored antimicrobial drugs for future use which is also intriguing and suggesting that in many cases previously prescribed antimicrobial therapy is not completed which is an additional factor contributing to the spread of antimicrobial resistance.

Reviewer 2 Report

This is an important study addressing a crucial issue to improve antimicrobial stewardship in the pediatric population. However, the following issue needs to be addressed before considering this manuscript for publication:

·       Figure 3: The authors reported in this figure cephalexin and Keflex as different types of antibiotics. However, Keflex is just the brand name of cephalexin. So, the reason for separating them is unclear. The same issue happened with Trimethoprim/Sulfamethoxazole and Bactrim as well. Moreover, they reported Valaciclovir (Valtrex) in this figure even though the study targets antibiotics and not antivirals.

Reviewer 3 Report

Author should update introduction section by incorporating certain information like how non-prescription use of antibiotics among children may lead to over or excessive antibiotic use and how this practice may contribute to AMR. 

Introduction section also need to provide rational specially in the second last paragraph that how this study would be different from already studies in the US? What is the research questions authors want to address and how?

As authors mentioned that they recruited study participants that visited children clinics. I assume that definitely their child would be sick before they planned to visit the clinic? if so did author collect this information? But if they were on routine visit then was it reported. Because, if their child was sick then definitely they were inclined towards antibiotics use by themselves. Please clarify what is the logic of recruiting such parents that were visiting these child clinics? 

Please provide the reference of statement '. In our previous study that included two safety-net clinics, 30% of 244 the respondents reported that they would use non-prescription antibiotics' in 'Study Setting, Sample Size, and Recruitment' heading.

Discussion is not up to the mark, only results were discuss, how these findings can be compared to the findings of others studies even from others countries. What are perspectives of parents while practices such non prescription use? . What are possible ways to reduce such practices?. Please incorporate all these.  

Reviewer 4 Report

Dear Authors,

I congratulate all the Authors for their contributions to the writing of the manuscript entitled “Prevalence and Predictors of Using Antibiotics without a Prescription in a Pediatric Population in the United States”.

I have few comments on the manuscript:

1.    Line 22: it’s unusual to start a sentence with number. Please rephrase this.

2.    Lines 24-26: n should be written in italics. For example, n = 69 in line 24.

3.    Line 38: a list of frequent antibiotics prescribed to children will be useful.

4.    Line 54: viral infections such as?

5.    Line 62: what are 219 and 153? n = 219?

6.    Line 76: Supplemental File S1 is missing. Please check this.

7.    Table 1: see above comment on how to write n. Please check the manuscript thoroughly.

8.    Figure 2: see above comment.

9.    Figure 3: see above comment.

10. Table 2: see above comment. p-value should be written in italics.

11.  Lines 177-179: younger caregivers were more likely to use antibiotics without a prescription. Why? Any thoughts on this?

12. Lines 218-231: this paragraph can be converted into Conclusion.

I consider the manuscript is sufficiently comprehensive and can be considered for publication in the Antibiotics journal after these issues have been properly addressed.

My sincere congratulations to all Authors.

Round 2

Reviewer 3 Report

Thank you to all the contributors to modify the manuscript as per my comments.